

# Indexing labeled sequences

Tatiana Rocher, Mathieu Giraud and Mikaël Salson

Université de Lille, CNRS, Centrale Lille, Inria, UMR 9189 CRIStAL, Lille, France

## ABSTRACT

**Background:** *Labels* are a way to add some information on a text, such as functional annotations such as genes on a DNA sequences. V(D)J recombinations are DNA recombinations involving two or three short genes in lymphocytes. Sequencing this short region (500 bp or less) produces labeled sequences and brings insight in the lymphocyte repertoire for onco-hematology or immunology studies.
**Methods:** We present two indexes for a text with non-overlapping labels. They store the text in a Burrows–Wheeler transform (BWT) and a compressed label sequence in a Wavelet Tree. The label sequence is taken in the order of the text (TL-index) or in the order of the BWT (TL$_{BW}$-index). Both indexes need a space related to the entropy of the labeled text.
**Results:** These indexes allow efficient text–label queries to count and find labeled patterns. The TL$_{BW}$-index has an overhead on simple label queries but is very efficient on combined pattern–label queries. We implemented the indexes in C++ and compared them against a baseline solution on pseudo-random as well as on V(D)J labeled texts.
**Discussion:** New indexes such as the ones we proposed improve the way we index and query labeled texts as, for instance, lymphocyte repertoire for hematological and immunological studies.

## INTRODUCTION

*Labels* are a way to add some information on a text, as the semantics of words on an English sentence or functional annotations such as genes on a DNA sequences. *Can we build an index that saves a labeled text like ACGCC . . . TTGA (of size 96), which have the label L$_1$ in the positions 12–41 and the label L$_2$ in the positions 56–96?* We consider here the case where a same label can be given to different (but similar) patterns and can occur several times in the text. We introduce two indexes which store a labeled text and answers to position–label association queries. Those indexes share some ideas with the RL-FMI (*Mäkinen & Navarro, 2004*) which uses a Burrows–Wheeler transform (BWT) and a Wavelet Tree (WT). Using a somewhat similar organization, we index a labeled text. The following sections present the TL- and TL$_{BW}$-indexes (text–label indexes) and their associated queries. The last section presents experimental results on simulated and real genomic data.

Let $T = t_0\ t_1\ \ldots\ t_{n-1}$ be a text of length $n$ over an alphabet of size σ. The text may be composed of several sequences, each sequence ending with the symbol $. Let $L = \{L_0, L_1\ \ldots\ L_{l-1}\}$ be a set of labels. A **labeled text** $(T, A)$ is here a text with non-overlapping labels: a letter should have at most one label. Each position $i$ of the text is labeled by exactly one label $a_i \in L \cup \{\varepsilon\}$, where the special ε label is put on every letter

Corresponding authors
Mathieu Giraud,
mathieu.giraud@univ-lille.fr
Mikaël Salson,
mikael.salson@univ-lille.fr

| 0 | 1 | 2 | 3 | 4 | 5 | 6 | 7 | 8 | 9 | 10 | 11 | 12 | 13 | 14 | 15 | 16 | 17 | 18 | 19 | 20 |
|---|---|---|---|---|---|---|---|---|---|----|----|----|----|----|----|----|----|----|----|----|
| A | A | C | A | G | C | $ | A | T | C | A | A | C | $ | A | G | C | T | T | T | $ |

$L_{1.2}$    $L_2$    $L_3$    $L_{1.1}$    $L_2$

**Figure 1** **The text $T$ = AACAGC\$ATCAAC\$AGCTTT\$, with three sequences, is labeled with the label string $A = L_{1.2}\ L_{1.2}\ L_{1.2}\ L_2\ L_2\ L_2\ \varepsilon\ L_3\ L_3\ L_3\ L_{1.1}\ L_{1.1}\ L_{1.1}\ \varepsilon\ L_2\ L_2\ L_2\ \varepsilon\ \varepsilon\ \varepsilon\ \varepsilon.$**

without label. $A = a_0\ a_1\ \ldots\ a_{n-1}$ is the label string. Figure 1 shows the text which will be used all over this article.

Given a **bit vector** $B = B[0]B[1]\ \ldots\ B[n-1]$, where each $B[i]$ is either 0 or 1, we define as $B[i, j]$ the vector $B[i]B[i+1]\ \ldots\ B[j]$. Let us call $rank(b, i, B)$ the number of times the bit $b \in \{0, 1\}$ appears in the prefix $B[0, i]$ and $select\ (b, j, B)$ the position $i$ of the $j$th appearance of the bit $b$ in $B$. Such a bit vector $B$ can be stored in $nH_0(B) + o(n)$ bits to support $rank$ and $select$ in $O(1)$, where $H_0(B)$ is the zeroth order entropy of $B$ (*Raman, Raman & Rao, 2002*).

The **BWT** (*Burrows & Wheeler, 1994*) is a reversible algorithm which reorganizes the letters of a text. The transformed text, $BWT(T)$, is the concatenation of the last letters of the lexicographically sorted rotations of the text. $BWT(T)$ is easier to compress and can be stored using $nH_k(T) + o(n)$ bits, where $H_k$ is the $k$th order empirical entropy of the text $T$. The **FM-index** (*Ferragina & Manzini, 2000*) uses the BWT, a table $C$, where $C[\alpha]$ gives the number of letters lexicographically smaller than $\alpha$, and a function $Occ(\alpha, i)$ giving the number of occurrences of $\alpha$ in $BWT(T)[0, i]$. The FM-index allows to efficiently search a pattern using backward search.

A **WT** (*Grossi, Gupta & Vitter, 2003*) is a binary tree storing a text, where each symbol from the alphabet corresponds to a leaf. The root is a bit vector where every position corresponds to the element it has to index. Any position marked 0 (respectively 1) corresponds to an element whose leaf is on the left (respectively right) descendant of the node. The process is repeated recursively until the leaves. For a text $A$ of length $a$ in an alphabet of size $l$, the construction of a balanced WT needs $O\left(a \lceil \log l / \sqrt{\log a}\, \rceil \right)$ time (*Munro, Nekrich & Vitter, 2016*) and requires $nH_0(A) + o(a \log l)$ bits when bit vectors are zero-order compressed (*Navarro & Mäkinen, 2007*).

The usual full-text indexes such as the FM-index (*Ferragina & Manzini, 2000*) or the LZ-index (*Kärkkäinen & Ukkonen, 1996*) do not index labeled texts. Some recent researches allow bidimensional range queries: *Arroyuelo et al. (2015)* represent an XML tree structure as a compressed bit vector and allow queries on structured patterns.

We focus here on *non-overlapping labels* and look for ways to efficiently store such labels along with sequences as well as to query them. The basic idea is that consecutive labels in $A$ are often the same: we will thus store *compressed label sequences*.

# MATERIALS AND METHODS

## TL-index: indexing labels over a text

Given a labeled text $(T, A)$, we define the TL-index as, using a FM-index built on a BWT $U$ to index the text, a bit vector $B_A$ marking the positions in the text where the labels change, and a WT $W_A$ indexing a compressed label sequence (Fig. 2A).

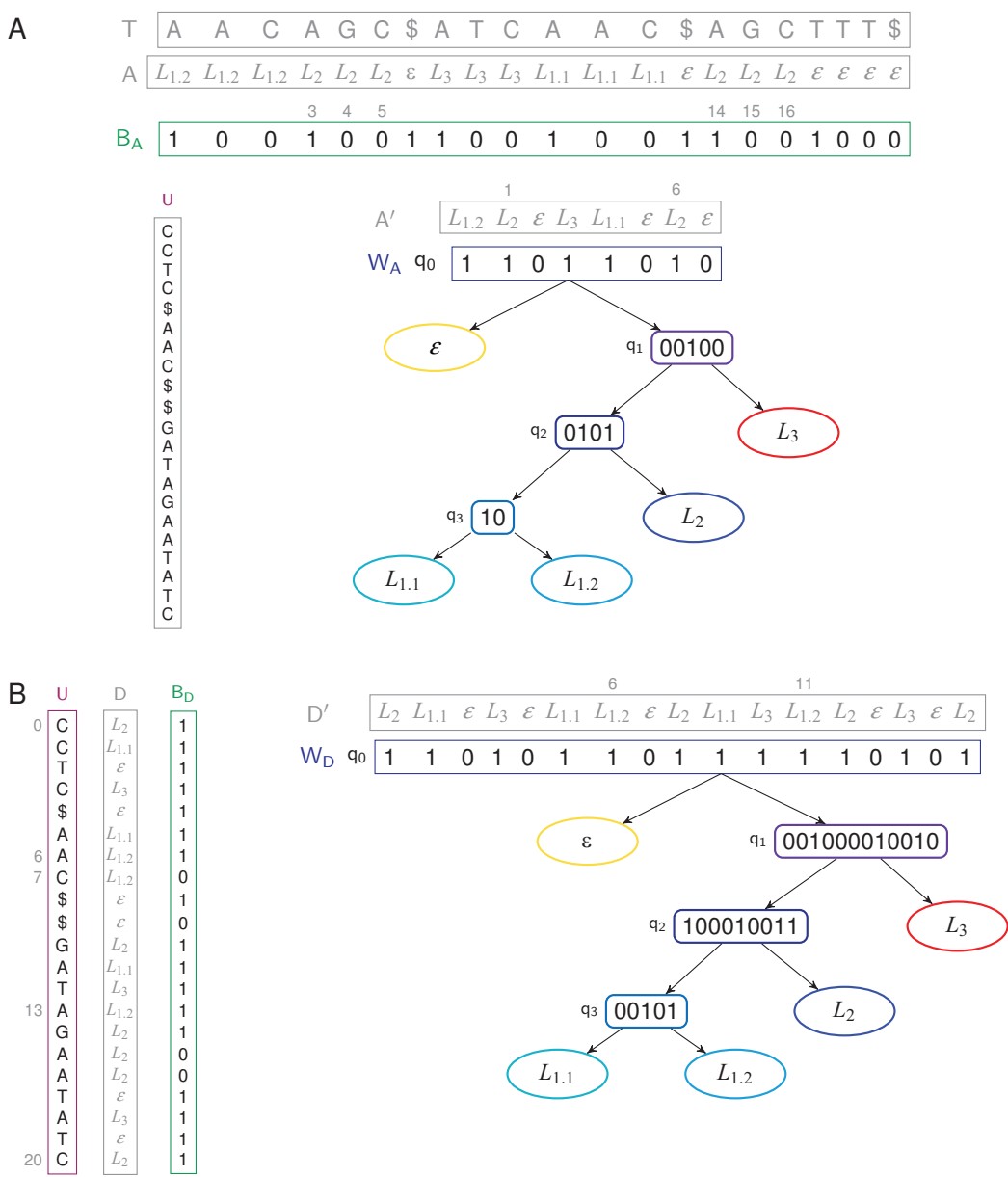

**Figure 2  TL- and TL$_{BW}$-indexes store a text of length 21 with four unique labels.** (A) TL-index. The WT $W_A$ has four internal nodes and five leaves. We have $B_A[4] = 0$ because $a_4 = a_3 = L_2$, and $B_A[3] = 1$ because $a_3 \neq a_2$. The label $a_3 = L_2$ is thus stored in $A'$, at position 1, hence $W\langle 1\rangle = a'_1 = L_2$. $A'$ has two occurrences of the label $L_2$: $W^{-1}\langle L_2\rangle = \{1, 6\}$, corresponding to the six positions $\{3, 4, 5, 14, 15, 16\}$ in $A$. (B) TL$_{BW}$-index. The root of the WT $D'$ is now built in the order of the BWT $U$. The WT $W_D$ has four internal nodes and five leaves. The label $d_6 = L_{1.2}$ is stored in $D'$, at position 6, hence $W\langle 6\rangle = d'_6 = L_{1.2}$. $D'$ has two occurrences of the label $L_{1.2}$: $W^{-1}\langle L_{1.2}\rangle = \{6, 11\}$, corresponding to the three positions $\{6, 7, 13\}$ in $U$. In both cases, the label sequences ($A$, $D$) and the compressed label sequences ($A'$, $D'$) are not stored in the indexes.

**BWT $U$.** Let $U = u_0 u_1 \ldots u_{n-1}$ be the BWT of $T = t_0 t_1 \ldots t_{n-1}$. As usually done, the FM-index samples every $\log^{1+\varepsilon} n$ values of a suffix array to retrieve the text positions of any occurrence (*Navarro & Mäkinen, 2007*).

**Bit vector $B_A$.** Let $B_A$ a compressed bit vector of length $n$ such that $B_A[0] = 1$, and, for $i \geq 1$, $B_A[i] = 0$ if $a_i = a_{i-1}$, and otherwise $B_A[i] = 1$.

**Wavelet Tree $W_A$.** Let $A' = \langle a_i \mid B_A[i] = 1 \rangle$. $A' = a'_0 \, a'_1 \, \ldots \, a'_{a-1}$ is called the *compressed label sequence*. It is a subsequence of $A$, of length $a$, containing only the changing labels according to the positions in $A$. The compressed label sequence $A'$ is stored using a WT $W_A$. $W_A$ is defined as $(Q, q_0)$, where $Q$ is the set of nodes of $W_A$ and $q_0 \in Q$ is the root node. Each node $q \in Q$ is $q = (q.val, q.label, q.left, q.right, q.parent)$, where $q.label \in L \cup \{\varepsilon\}$ is the label of $q$ and $q.parent$ is the parent node of $q$ ($q_0.parent$ being the null value $\perp$). Both $q.left$ and $q.right$ are child nodes in $Q \cup \{\perp\}$. A *leaf* is a node $q$ where $q.right$ and $q.left$ are $\perp$. Each leaf $q$ is linked to a label $q.label \in L \cup \{\varepsilon\}$. The $l + 1$ leaves are exactly the labels of $L \cup \{\varepsilon\}$ and we define $leaf(q.label) = q$. On a leaf $q$, we have $q.val = \perp$. Let $q$ be a non-leaf node: we have $q.label = q.val$ is the bit vector rooted at $q$ in the WT. We explain in "Shaping the WT for a Label Hierarchy" how $W_A$ can be further shaped depending on a label hierarchy.

$W_A$ is part of the index. This WT is used to answer efficiently bidimensional range queries where the labels and the label positions are the two dimensions (*Mäkinen & Navarro, 2007*). A balanced WT has a height of $\log l$, with $l$, the number of leaves. The accessor $W\langle i \rangle$ returns $a'_i$ in $O(\log l)$ time. This is a classical query within a WT. Given a label $L_x \in L$, the function $selectW\langle L_x, i \rangle$ gives the position of the $i$th $L_x$ label in $A'$ in $O(\log l)$ time. The accessor $W^{-1}\langle L_x \rangle$ gives the list of positions in $A'$ where $a'_i = L_x$. It runs in $O(\log l \times occ)$ time, with $occ$ the number of occurrences of $L_x$ in $A'$.

## TL$_{BW}$-index: indexing labels in the order of the BWT

The BWT tends to store text repetitions consecutively. As those repetitions may have the same labels, it would be interesting that the labels benefit from the BWT reordering. Hence, labels can also be stored in the order of $U$.

Given a labeled text $(T, A)$, the TL$_{BW}$-index is defined as $(U, B_D, W_D)$ (Fig. 2B). The BWT $U$ is built in the same way as the TL-index. Let $D = d_0 \, d_1 \, \ldots \, d_{n-1}$ the labels in the order of $U$. The bit vector $B_D$ of size $n$ is such that $B_D[0] = 1$, and, for $i \geq 1$, $B_D[i] = 0$ if $d_i = d_{i-1}$, and otherwise $B_D[i] = 1$. Let $D' = \langle d_i \mid B_D[i] = 1 \rangle$. $D' = d'_0, d'_1 \ldots d'_{d-1}$ is a compressed label sequence of length $d$, subsequence of $D$. The WT $W_D$ now indexes the compressed label sequence $D'$. The TL$_{BW}$-index will be slower to build than the TL-index as it needs $D$. On the other side, as it is aware of the order of letters in the BWT, it will be able to support faster text/label queries.

## Queries

The indexes allow the following classical queries.

- *label($t_i$)* (also called *access($a_i$))*—**Which label is on the letter $t_i$?** This query is done is $O(\log l)$ time in the TL-index, and in $O(\log^{1+\varepsilon} n + \log l)$ time in the TL$_{BW}$-index since it has to convert positions in $U$ order (see Fig. 3).

- *findP(P)* **P—Which are the occurrences of a pattern $P$?** It is solved with the FM-index alone (*Ferragina & Manzini, 2000*). This query runs in $O(|P| + occ \times \log^{1+\varepsilon} n)$ time in both indexes, where $occ$ is the number of occurrences of $P$ in $T$.

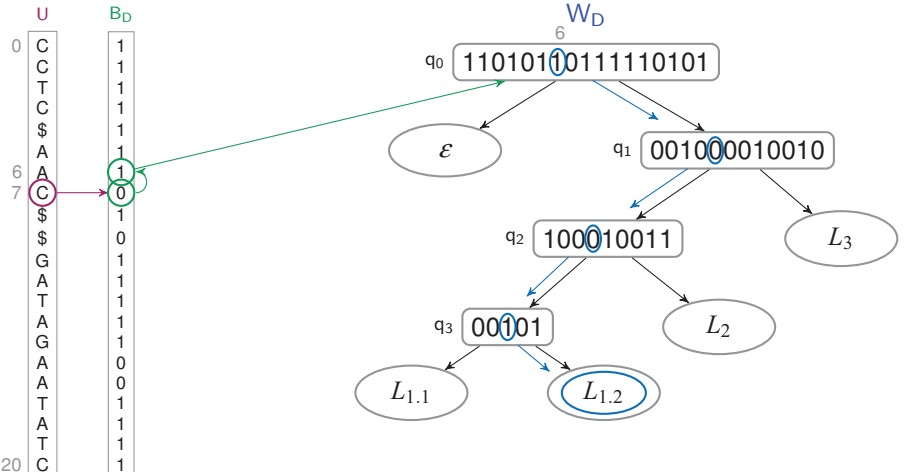

**Figure 3 Finding the label of a letter in a $TL_{BW}$-index.** The letter $u_7$ corresponds to a 0-bit in $B_D$. The previous 1-bit in $B_D$ is the bit at position 6. It's the 7th 1-bit of $B_D$, it corresponds to the bit at position 6 in $W_D$'s root, which label is $W\langle 6 \rangle = L_{1.2}$.

- **findL($L_x$)—Which text positions are labeled $L_x$?** The query runs in $O(y \times \log l)$ time in the TL-index, and in $O(y(\log l + \log^{1+\varepsilon} n))$ time in the $TL_{BW}$-index, where $Y = W^{-1}\langle L_x \rangle$ and $y = |Y|$. See Fig. 4.

The three previous queries are well known in text indexes. The two next queries search for a pattern and a label at the same time.

- **countPL($P, L_x$)—How many text positions are labeled $L_x$ and begin an occurrence of a pattern $P$?** As in the *findL($L_x$)* query, the occurrences of $P$ are found in $U$, in the positions from $i$ to $j$.

  TL-index: We translate all these *occ* occurrences to have the corresponding positions in the text. For each of them, we run the query *label($t_i$)*. The total time is $O(|P| + occ (\log n^{1+\varepsilon} + \log l))$.

  $TL_{BW}$-index: See Algorithm 1. $i$ and $j$ correspond to the positions $i' = rank(1, i, B_D)$ and $j' = rank(1, j, B_D)$ in the root $q_0$ of $W_D$. We then use an accessor customized from the *rangeLocate* function of (*Mäkinen & Navarro, 2007*), simulating a two-dimensional range query on $[L_x, L_x] \times [i', j']$ in $W_D$ to give the set of positions $Z = \{z \mid a'_z = L_x$ and $i' \le z \le j'\}$ in $O(|Z| \times \log l)$ time. This accessor first traverses the WT from the root to the leaf $L_x$ and then traverses it back to the root to find the positions in $q_0.val$ which match the label in the given range. For every position found, we find the corresponding positions in $B_D$ and expand them to the following 0-bits in $B_D$. This query runs in $O(|P| + |Z| \times \log l)$ time.

- **findPL($P, L_x$)—Which text positions are labeled $L_x$ and begin an occurrence of a pattern $P$?**

  TL-index: This query is exactly the same as *countPL($P, L_x$)*.

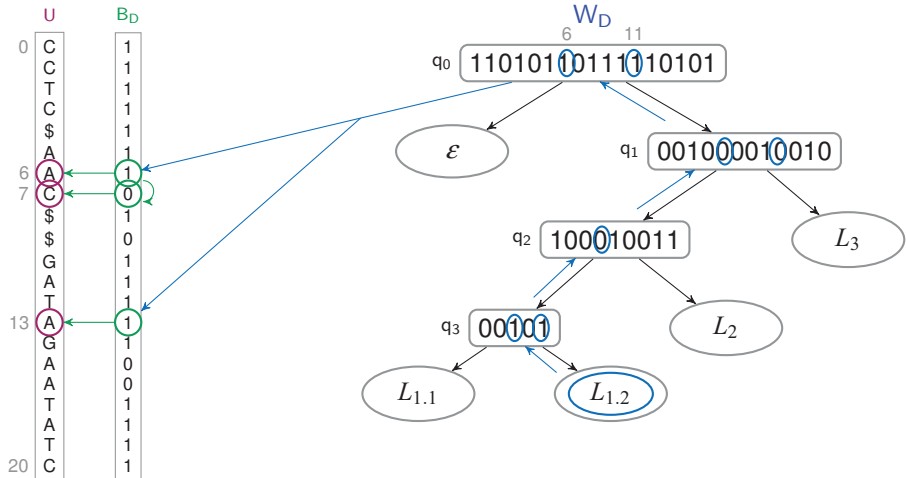

**Figure 4 Finding the letters which have a $L_{1.2}$ label in a TL$_{BW}$-index.** $Y = W^{-1}\langle L_{1.2}\rangle = \{6, 11\}$. For every bit of $Y$, we access to the 1-bit corresponding in $B_D$ ($\{6, 13\}$), and all the 0-bits which follow it ($\{7\}$). The corresponding letters of $U$ are labeled $L_{1.3}$, at positions $\{6, 7, 13\}$.

---

**Algorithm 1** *countPL($P$, $L_x$)*: Count the positions starting a pattern $P$ and labeled with $L_x$

$(i,j)$ from *findP($P$)*  ▷ starting and ending positions of occurrences of $P$ in $U$

$i' = rank(1, i, B_D)$

$j' = rank(1, j, B_D)$

$C = path(L_x)$  ▷ bit vector representing the path from the root to *leaf($L_x$)*

$node = q_0$

**for** $p$ in 0 to $|C| - 2$ **do**  ▷ loop corresponding to *rangeLocate* in (*Mäkinen & Navarro, 2007*)

  $i' = rank(C[p], i' - 1, node.val)$

  $j' = rank(C[p], j', node.val) - 1$

  $node = (C[p] == 0)?\ node.left : node.right$

  **if** $i' > j'$ **then return** 0

$cnt = 0$

**for** $k$ in $i'$ to $j'$ **do**

  $k' = selectW\langle L_x, rank(C[|C| - 1], k, node.val)\rangle$  ▷ *i*th $L_x$ label in $A'$

  $i'' = select(1, k', B_D)$

  $j'' = select(1, k' + 1, B_D)$

  $cnt = cnt + j'' - i''$  ▷ positions $[i'', j'' - 1]$ in $U$ taken into account

**return** $cnt$

---

TL$_{BW}$-index: We use the *countPL($P$, $L_x$)* query detailed in Algorithm 1, replacing the counter *cnt* with a list $Y$ holding the matching positions in $U$. The positions are converted at the end in the text order. This query runs in $O(|P| + |Z| \times \log l + y \times \log^{1+\varepsilon} n)$ time, with $|Y| = y$.

As $|Z| \leq y \leq occ$, the *countPL()* and the *findPL()* queries may thus be faster on the $TL_{BW}$-index, $|Z|$ depending of the compression $B_D$ can do on $Y$.

Note that the *countPL(P, L$_x$)* query could be faster if the WT was directly built on the labels, without the intermediate bit vector $B_D$ ($B_A$): the answer would be known while reaching the leaf of the WT in $O(|P| + \log l)$ time. We chose to favor the execution time of the *findPL(P, L$_x$)* query, as well as the size of the structure (when $A$ can be compressed).

The *findPL(P, L$_x$)* can vary to find the patterns $P$ which have the label $L_x$ on the $i$th position of the pattern, or in any pattern's position. Adapting this queries is easy in the TL-index as we find the positions of the pattern in the BWT, translate them in the text order and then find the label of the $i$th position following each of them. In the $TL_{BW}$-index, we find the patterns' positions in the BWT, access to the $i$th letter (we need to sample the BWT to read the patterns in the forward way), and find the label as usual. To have the label in any pattern's position, in the TL-index we need to find a label $L_x$ between the first and last letter of the pattern (with only two access in the WT) but in the $TL_{BW}$-index, we look for the label of all the pattern's letters.

## Construction and space

We recall that the text $(T, A)$ is of length $n$ and is labeled with $l$ unique labels. As defined above, the indexes store $U$ in $nH_k(T) + o(n)$ bits. The TL-index stores the bit vector with *rank* and *select* capabilities in $nH_0(B_A) + o(n)$ bits. The size of $W_A$ depends on the compressed label sequence $A'$, of length $a$. $W_A$ takes $a\, H_0(A') + o(a \log l)$ bits. Similarly, the $TL_{BW}$-index stores $B_D$ in $nH_0(B_D) + o(n)$ bits and $W_D$ takes $d\, H_0(D') + o(d \log l)$ bits, where $d$ is the length of $D'$. The BWT can be built in linear time while using little space (*Belazzougui et al., 2016*; *Munro, Navarro & Nekrich, 2017*). $B_A$ is built while reading $A$ in $O(n)$ time. To make $B_D$, we need to read the labels in the order of the original data file in $O(n)$ time. To make $W_A$, we find the occurrence of each label, corresponding to a 1-bit in $B_A$, in $O(a)$ time. Then we form the shape of $W_A$ in $O(l)$. The labels corresponding to a 1-bit are extracted to make the WT's root $q_0$. For each node containing at least two labels, we separate them by following the shape previously calculated, in $O(a \lceil \log l / \sqrt{\log a} \rceil)$. We build $W_D$ the same way. The TL-index has thus a size of $nH_k(T) + nH_0(B_A) + a\, H_0(A') + o(n \log l)$ bits, assuming $\sigma = O(l)$, and is built in $O(n + l + a \lceil \log l / \sqrt{\log a} \rceil)$ time. The $TL_{BW}$-index has a size of $nH_k(T) + d\, H_0(D') + o(n \log l)$ bits and is built in $O(n + l + d \lceil \log l / \sqrt{\log d} \rceil)$ time.

## Shaping the WT for a label hierarchy

Labels may be further organized into a *label hierarchy*, given an additional set $F = \{F_0, \ldots, F_{f-1}\}$ of *label families* (Fig. 5A). Both TL- and $TL_{BW}$-indexes can be adapted: The WT $W$ (either $W_A$ or $W_D$) will be shaped according to the hierarchy, and internal nodes $q$ of $W$ may have non-empty $q.label$ values belonging to $F$. For example, in Fig. 5B, one can set on either index $q_1.label = L_1$, where $L_1$ is the label family gathering $L_{1.1}$, $L_{1.2}$ and $L_{1.3}$.

The *findL()* and *findPL()* queries naturally extend to label families. With the hierarchy depicted in Fig. 5, *findL(L$_1$)* has to find the positions that have a label $L_{1.1}$, $L_{1.2}$ or $L_{1.3}$.

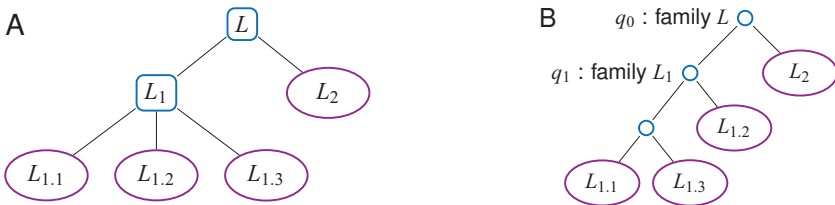

**Figure 5 A label *n*-ary hierarchy (A) can be represented with a binary tree shaping the Wavelet Tree (B).** The label family $L_1$ has here three descendants, $L_{1.1}$, $L_{1.2}$ and $L_{1.3}$.

Such a query does not need to be iterated on every label of the family $L_1$, but rather directly starts at the corresponding internal node ($q_1$ on Fig. 5B).

Shaping $W$ for a label hierarchy may increase the height $w$ of the WT to $O(l)$ in the worst case. To have a lower space consumption and a faster average query time, one can give a Huffman shape to $W$ (*Huffman, 1952*). A leaf which corresponds to a frequently used label will be higher in the tree than a leaf of a rarely used label. Depending on the label hierarchy, the average height of the tree is $H_0(A')$ in the best case while in the worst case it could be $O(l)$. If no label hierarchy is given, the average height $w$ will be $H_0(A')$ (*Mäkinen & Navarro, 2005*).

# RESULTS AND DISCUSSION

## HT-index: a baseline index

We compared the TL- and $TL_{BW}$-indexes with a baseline solution called HT-index, indexing the text $T$ with a BWT. The labels are stored in a map linking each label to the list of its (*start,end*) positions. We also store the labels in the text order with the compressed bit vector $B_A$ and, stored in plain form, $A'$. This enables the *findL*($L_x$) query in $O(y)$, where $Y'$ is the list of pairs (*start,end*) which represent the occurrences and $y = |Y'|$. Note that $Y$ (in the TL-index) and $Y'$ represent the same information as the labels are stored in the text order in both indexes. The *label*($t_i$) query runs in $O(1)$ time. This solution is not space-efficient with repeated labels: it needs $nH_k(T) + nH_0(B_A) + a + l' + o(n)$ bits, where $a$ is the size of $A'$ and $l'$ the number of labeled factors. The query times are summarized in Table 1.

## Evaluation procedure

The three indexes were implemented in C++. We used the SDSL-Lite library (*Gog et al., 2014*) to build the bit vectors and the WT. We used the RopeBWT2 library (*Li, 2014*), which builds a BWT in $O(n \log n)$ time on small DNA sequences, as it is very efficient for storing and appending sequences corresponding to our projected application (*Cox et al., 2012*). As RopeBWT2 does not sample the suffix array, we iterate over the text until we find a $ symbol. To have results close to the usual FM-index sampling in $O(\log^{1+\varepsilon} n)$ steps, we use sequences of length 50, which is similar to the sampling distance usually chosen in practice. The real files have longer sequences, thus longer sampling distances. The queries relying on the BWT will be slower and therefore cannot be compared between real and simulated files. We build the $B_A$ (or $B_D$) bit vectors, compress them using the `rrr_vector`

**Table 1 Query time complexities for HT-index, TL-index and TL$_{\text{BW}}$-index.**

| Requests | HT-index | TL-index | TL$_{\text{BW}}$-index |
|---|---|---|---|
| $label(i)$ | $O(1)$ | $O(\log l)$ | $O(\log^{1+\varepsilon} n + \log l)$ |
| $findP(P)$ | | $O(|P| + occ_P \times \log^{1+\varepsilon} n)$ | |
| $findL(L_x)$ | $O(y)$ | $O(y \times \log l)$ | $O(y \times (\log l + \log^{1+\varepsilon} n))$ |
| $countPL(P, L_x)$ | | | $O(|P| + |Z| \times \log l)$ |
| $findPL(P, L_x)$ | $O(|P| + occ_P \times \log^{1+\varepsilon} n)$ | $O(|P| + occ_p \times (\log^{1+\varepsilon} n + \log l))$ | $O(|P| + |Z| \times \log l + y \times \log^{1+\varepsilon} n)$ |

**Note:**
Note that we have $|Z| \le y \le occ_P$. The $label(i)$ and $findL(L_x)$ queries are faster in the HT-index and the TL-index as the HT-index needs a sampling time. However, the $countPL(P, L_x)$ and $findPL(P, L_x)$ are faster in the HT-index.

class of SDSL-Lite, and finally build $W_A$ (or $W_D$) using a shape we added in SDSL-Lite, which depends of the label hierarchy.

We evaluated the build time, the index size, as well as the run time of three of the queries detailed in "Queries"—$findP(P)$ behaving similarly in all indexes and $countPL(P, L_x)$ being very similar to $findPL(P, L_x)$. The three indexes were tested on various datasets of labeled texts, each with 100 M characters (Table 2). Datasets and code are available at http://www.vidjil.org/data/#2018-peerjcs.

- **Simulated files with random sequences and random labels.** All sequences and labels are random ($d \sim 0.8n$).

- **Simulated files, with random sequences but fixed labels.** Here a given label is associated to the same pattern, possibly with some variation, and we alter the proportion of labeled letters (5–100%), the variation in the label's pattern (0–50%, more variations giving random labels), the number of unique labels (10–1,000), the length of the labels (5–100 letters). The dataset has 546 files, two of those files are shown in Table 2.

- **Genomic sequences with immunologic labels.** A person's immune system can be described with a set of labeled DNA sequences with V(D)J recombinations. The dataset, detailed below, uses 838 K labels from 355 unique labels, with $d \sim 0.26n$.

## A dataset of DNA sequences with immunologic labels

The adaptive immunity is the mechanism thanks to which our body defends itself against infections. When B and T-cells, or *lymphocytes*, are under maturation, their DNA undergo a recombination which allows several billions possibilities from a register of a thousand genes (*Tonegawa, 1983*). For example, the V(D)J recombination `V4*02 4/ACGT/0 J1*12` means that the DNA sequence is composed from the `V4*02` gene without the four last letters, then the `ACGT` sequence, then the `J1*12` gene.

A person's immune system can thus be described with a set of labeled DNA sequences encoding V(D)J recombinations (Fig. 6). These sequences can be sampled by next-generation sequencing with bioinformatics analysis (*Bystry et al., 2016*, *Duez et al., 2016*).

The tested DNA sequences come from patients 09, 12, 14 and 63 from a public dataset on a study on acute lymphoblastic leukemia (*Salson et al., 2017*). They have 100 M letters

**Table 2 Size, build and query times of three indexes indexing labeled texts, on three simulated files and on a genomic DNA sequence file with immunologic labels.**

|  | Random | | | Fixed #1 | | | Fixed #2 | | | Genomic | | |
|---|---|---|---|---|---|---|---|---|---|---|---|---|
| Sequence size | 50 | | | 50 | | | 50 | | | 264 | | |
| Lab. ($t/u$) | 3.7 M / 1,000 | | | 4 M / 100 | | | 4 M / 10 | | | 852 K / 355 | | |
| Lab. avg size | 12.6 | | | 25 | | | 5 | | | 110.3 | | |
| Lab. letters (%) | 47 | | | 100 | | | 20 | | | 92 | | |
| Variation (%) | 100 | | | 5 | | | 50 | | | ?? | | |
| $a = \ldots/d = \ldots$ | $0.06n/0.77n$ | | | $0.06n/0.16n$ | | | $0.08n/0.36n$ | | | $0.016n/0.26n$ | | |
|  | TL | TL$_{BW}$ | HT | TL | TL$_{BW}$ | HT | TL | TL$_{BW}$ | HT | TL | TL$_{BW}$ | HT |
| Size (MB) | **104** | 184 | 217 | **33** | 38 | 114 | **99** | 116 | 209 | **13** | 33 | 35 |
| Time (s) | **18** | 80 | **15** | 13.2 | 77 | **11** | 14.6 | 73.8 | **13.2** | **10.2** | 65 | **10.2** |
| $label(t_i)$ (μs) | 3.84 | 21.54 | **0.73** | 4.50 | 20.02 | **0.55** | 1.22 | 21.2 | **0.67** | 2.98 | 81.1 | **0.37** |
| $findL(L)$ (μs/l) | **0.4** | 34 | **0.4** | **0.12** | 13.1 | 0.20 | 0.40 | 21.41 | **0.21** | **0.04** | 52.5 | 0.17 |
| $findPL(P, L)$ (s) | 34.7 | **5.13** | 29.7 | 26.0 | **3.19** | 20.9 | 30.64 | **3.00** | 28.91 | 131.1 | **3.83** | 127.8 |

**Notes:**
All files have 100 M characters, and differ by the number of total and unique labels ("Lab (t/u)") and their size ("Lab. avg size"), by the ratio of labeled letters ("Lab. letters"), and by the variation between sequences labeled by the same label ("Variation"). Queries use patterns $P$ with three letters. Times were averaged on 1 M launches ($label()$) or at least five launches (other queries). Times for $findL(L)$ are reported per letter. Best or close-to-the-best results are in bold.

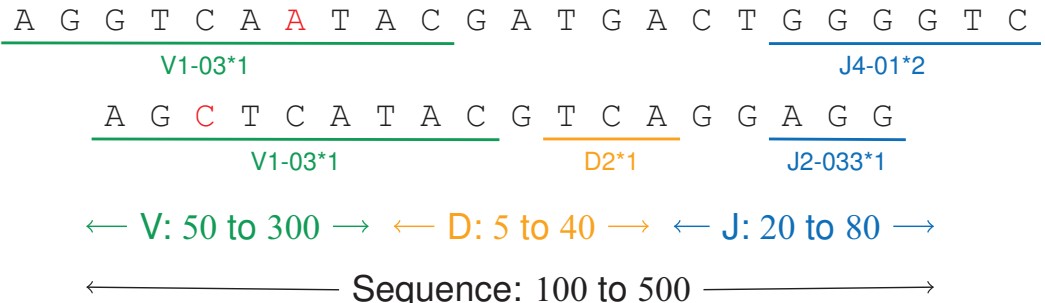

**Figure 6 V(D)J recombinations.** The first sequence is an immunoglobulin "light chain," that is a VJ recombination with two labels (one V gene, positions 0–9, and one J gene, positions 17–22). The second sequence is a "heavy chain," that is a V(D)J recombination.

and 838 K labels from 355 unique labels, making a 117 MB file. Each DNA sequence has between 100 and 350 letters and two or three labels, each label having a size between 5 and 200 letters (Fig. 7). For a given label, the labeled subsequences may vary up to 15% due to amplification and sequencing errors.

## Results

**Index sizes.** The Table 1 shows the results. As expected, the size of $U$, $B$ (either $B_A$ or $B_D$) and $W$ (either $W_A$ or $W_D$) grows linearly with the number of indexed elements (data not shown). The TL-index is the smallest index, and the TL$_{BW}$-index is generally slightly larger. The compression is directly related to $a$ and $d$. The file with random labels ($d = 0.77n$) is hard to compress, whereas the files with a low $d/n$ ratio give a 2× to 7× compression. Figure 8 further shows how these sizes vary: As expected, the indexes are larger when there are less consecutive identical labels in $T$ or in $U$, thus more 1-bits in $B$.

```
>TRGV2*02:1-177 TRGJ1*01:192-226
GGAAGGCCCCACAGCGTCTTCAGTACTATGACTCCTACAACTCCAAGGTTGTGTTGGAA (...)

>IGHV3-11*01:1-251 IGHD6-19*01:260-279 IGHJ5*02:285-331
GGAGGTCCCTGAGACTCTCCTGTGCAGCCTCTGGATTCACCTTCAGTGACTACTACATG (...)
```

**Figure 7 Two annotated sequences from the dataset.**

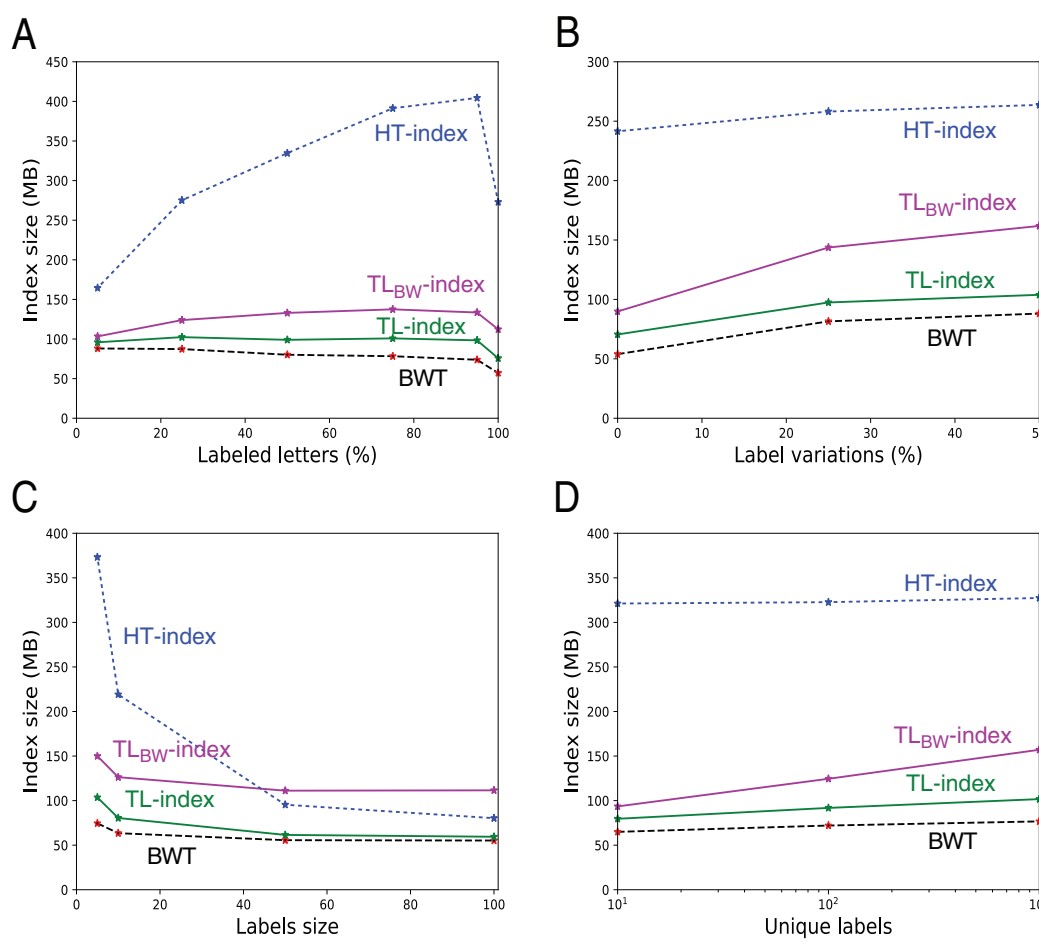

**Figure 8 Size of the indexes and size of the underlying BWT in the 546 files with fixed labels.** The additional size in TL- and TL$_{BW}$-indexes mostly depends from the size of the compressed label sequences $A'$ and $D'$. They grow when there are more labeled letters (A), more variation in the labels (B), or when the number of distinct labels increase (D). Note that when all the letters are labeled (A, 100%), there is a small decrease in the index size because there is no random letters between the patterns. The indexes shrink when the labels grow (C), as there are more common suffixes in the label sequences.

Note that when there are more labeled letters, the text is more similar (as labels are placed on similar substrings), hence a decrease in the BWT size (A, C). $W$ increases while the number of unique labels increases (D), the height of $W$ increasing logarithmically with the number of unique labels.

**Build time.** Most of the build time of TL$_{BW}$-index is devoted to build $D'$. For TL-index, the building of $U$ takes most of the total time.

**Queries.** The *label*() query is faster on the HT-index. As expected, the $TL_{BW}$-index needs more time for *label*() and *findL*(), as it needs to translate positions from the BWT back to the text. Note that locating the positions in the text takes about the same time as *label*($t_i$) in the TL-index. However, for the complex *findPL*($P, L$) query, the $TL_{BW}$-index is the fastest solution because the position translation is only done on the letters which have both the label and the pattern. For the TL-index and HT-index, the actual time of the *findPL*($P, L$) queries is more affected by the number of pattern occurrences than the number of final occurrences (between 0 and 100 K depending on the file).

On the genomic dataset, the sequences are longer: The $TL_{BW}$-index suffers here even more from the missing suffix array sampling of the implementation for queries *label*() and *findL*(). However, on the *findPL*($P, L$) query, the other indexes are penalized due to the sparsity of the sampling, bringing a more than $30\times$ difference with $TL_{BW}$-index.

## CONCLUSION

The TL-index and $TL_{BW}$-index store a labeled text and allow efficient queries. They can be constructed from files of some MB in a few seconds. Experiments confirm that the indexes stay small when the text is redundant (thus a smaller $U$), when each label describes a pattern with few variations (many 0-bits in $B$, thus a smaller $W$), and when few letters are labeled (thus a small $W$). However, the TL-index and $TL_{BW}$-index are robust even to non-redundant text with almost random labels everywhere. The $TL_{BW}$-index needs more memory space than the TL-index but is more efficient in combined label/pattern queries. Those structures might be used on any labeled data, such as DNA sequences, but also on natural language texts or on music sheets with some semantic annotations.

Perspectives include improvement of the implementation, with label families queries or parameterizing the distance between samples in the FM-index to offer a space-time trade off. Within SDSL we could use the `sd_vector` bit vector instead of the `rrr_vector` bit vector which should improve space consumption when the bit vectors are very sparse. However, this would only minimally improve the global space consumption of the index. We plan to use one of the indexes in a *clone database* for hematological malignancies: It will allow comparison of V(D)J recombinations between different samples of a patient or between several patients.

## ACKNOWLEDGEMENTS

We thank anonymous reviewers for their insightful comments on earlier versions of this article, as well as the EuroClonality-NGS consortium for insightful discussions.

### Funding

This work was supported by Université de Lille, SIRIC ONCOLille, and Région Hauts-de-France. The funders had no role in study design, data collection and analysis, decision to publish, or preparation of the manuscript.

## Grant Disclosures

The following grant information was disclosed by the authors:
Université de Lille, SIRIC ONCOLille, and Région Hauts-de-France.

## Competing Interests

The authors declare that they have no competing interests.

## Author Contributions

- Tatiana Rocher conceived and designed the experiments, performed the experiments, analyzed the data, prepared figures and/or tables, performed the computation work, authored or reviewed drafts of the paper.
- Mathieu Giraud conceived and designed the experiments, analyzed the data, authored or reviewed drafts of the paper.
- Mikaël Salson conceived and designed the experiments, analyzed the data, authored or reviewed drafts of the paper.

## Data Availability

Code and data are available as Supplemental Dataset Files and from http://www.vidjil. org/data/#2018-peerjcs.

## Supplemental Information

Supplemental information for this article can be found online at http://dx.doi.org/ 10.7717/peerj-cs.148#supplemental-information.

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
