# Peer review of "Indexing labeled sequences"

_PeerJ Computer Science, doi:10.7717/peerj-cs.148_

## Round 0.1 · original submission · Minor Revisions

Dear authors,

We are happy to see that your manuscript has been looked upon favorably by all the reviewers. Overall consensus is that this manuscript only requires minor revisions. While most of the minor revisions are immediate, there are a couple which may need a little more thought and restructuring.

We look forward to your revised manuscript.

best regards,
Rahul Shah

Reviewer 1 ·

Basic reporting

This is an easy to read article targeted to wider audience. The article illustrates how easy it is to design a tailored solution, exploiting modern compressed text indexing, to combine labels and DNA sequence into a common data structure that allows to make useful queries on the content. All the components are widely known, but this combination has not been previously reported; there is, though, an even more broad generalization of adding an XML tree on top of a sequence [1], which encompasses linear labelling as a special case. That article is not targeted to a general audience, so reporting this special case with a tailored (and different) solution is completely desirable.

English should be improved. I spotted some very easy typos detectable by a spell checker, and with high probability there are more (page 1, line 43: let call -->let us call; page 6, line 137: anwser --> answer).

[1] Diego Arroyuelo, Francisco Claude, Sebastian Maneth, Veli Mäkinen, Gonzalo Navarro, Kim Nguyen, Jouni Sirén, Niko Välimäki. Fast in-memory XPath search using compressed indexes. Softw., Pract. Exper. 45(3): 399-434 (2015)

Experimental design

As this is a new problem, there is no good baseline to compare. Comparison to naive solution is fine.

Validity of the findings

Authors show a proof of concept with a biological question motivating the study.

Cite this review as

Reviewer 2 ·

Basic reporting

The authors essentially consider the problem of indexing a pair of strings $T \in \Sigma*$ and $L \in {1, \ldots, \ell}$ such that, given a pattern $P$ and a integer $x$ between 1 and $\ell$, they can quickly count and/or return all the positions $j$ such that $T [j..j + |P| - 1] = P$ and $L [j] = x$.

Their solution is essentially to store an FM-index for $T$ together with a wavelet tree for the permutation $W$ of $L$ in which $W [j] = L [\SA [j]]$, where $\SA$ is the suffix array of $T$. Given $P$ and $x$, they use the FM-index to find the suffix array interval for $P$, then use the wavelet tree to count and/or find the occurrences of $x$ in that interval in $W$, then possibly use the FM-index's suffix-array sample to find also the corresponding positions in $T$.

The solution itself is correct, but obvious. The explanation is unnecessarily complicated, some of the bounds cited are not the best known, and several of the references are outdated. I would suggest the authors significantly tighten up their presentation before publication, although none of the problems are critical.

Experimental design

No comment.

Validity of the findings

The findings are correct, although not profound.

Additional comments

It seems fairly easy to build an $O (n \log n)$-bit index such that, given a pattern $P$, a label $x$ and a position $i$ in $P$, we can reasonably quickly find all the positions where $P$ occurs in $T$ with its $i$th character labelled $x$. Offhand, however, I don't see how to reduce the space to $O (n \log \Sigma)$. A solution to that problem might make the article more interesting for researchers familiar with pattern matching.

Cite this review as

Reviewer 3 ·

Basic reporting

This paper presents two indexes for labels texts. Each position of a text T of length n is marked with exactly one (possible empty) label.
The label string A is therefore also of length n.

The first index uses an FM-index over T and a WT over the run-length compressed version of A. The WT stores the sequence of run heads, while bit vector B_A marks the first position of the runs in T.
Since a search in the FM-index results in a SA range, access to A is not cheap since SA positions have to be translated into text positions. The cost of this translation is proportional to the SA-sampling s_SA.

In a second version of the index, A is also transformed in SA-order. Run-length compression is again applied.

A set of operations is defined and the authors show how to answer these operations. The algorithms are compositions of already known results.36: on an alphabet -> over an alphabet

Experimental design

The experimental section compares an implementation of the two indexes to a naive baseline. The implementation is based on the SDSL library but important details, i.e. which SA sampling strategy was used, are missing in the current version of the article. It is also unclear why the author opted for rrr_vector to represent the bit vectors. It is expected that sd_vector is superior to rrr_vector for long labels.

I suggest to fix the last two issues and accept the paper.

Validity of the findings

The index is a composition of already know techniques and its practical implementation and the experiments are worth a publication. The benchmark is available and the experiments seem to be sound.

Additional comments

Details:
40: The figure 1 -> Figure 1
58: there is a better construction of the WT (Munro, Nekrich, Vitter TCS 2016); ok mentioned in 146
71: The usual sampling is log^{1+\epsilon} n in theory. Please consider also practice: Gog and Navarro (SEA 2014) present a practical way to sample SA and ISA at the same time. Ferragina, Siren, and Venturini (ESA 2011) experimented with distribution-aware sampling.
Figure 2: These TL -and .. -> TL- and
106: figure 3 -> Figure 3, also 112
166: Is is not necessary to keep the ordering. This is not possible with Huffman but with Hu-Tucker codes.
221: The Figure -> Figure

Cite this review as

---

## Round 0.2 · accepted · Accept

Thank you for all your work in revising your manuscript. The changes you incorporated have definitely enriched the manuscript.

---

## Author Rebuttal · Round 0.2

**Response to the reviewers** – *Indexing labeled sequences*

Dear Editor,

We thank you, and the reviewers, for your positive appreciations on our manuscript #CS-2017:09:20302:1:0:NEW entitled *"Indexing labeled sequences"*.

# Reviewer 1

*There is [. . . ] an even more broad generalization of adding an XML tree on top of a sequence [1], which encompasses linear labelling as a special case. That article is not targeted to a general audience, so reporting this special case with a tailored (and different) solution is completely desirable.*

We appreciate this valuable comment. We integrated the reference at the end of the introduction to give a wider perspective.

# Reviewer 2

*Some of the bounds cited are not the best known, and several of the references are outdated.*

We are not sure what bounds and references the reviewers had in mind. However we noticed that a 2016 paper by Munro *et al* improved the wavelet tree construction. We have integrated it to the paper.

*It seems fairly easy to build an $O(n \log n)$-bit index such that, given a pattern $P$, a label $x$ and a position $i$ in $P$, we can reasonably quickly find all the positions where $P$ occurs in $T$ with its ith character labelled $x$. Offhand, however, I don't see how to reduce the space to $O(n \log \sigma)$. A solution to that problem might make the article more interesting for researchers familiar with pattern matching.*

We are grateful to the reviewer for this interesting comment. We recall that our approach is in $O(n \log \sigma)$ bits. We added a discussion on how we could slightly modify our approach to search the label at another position than the first one of the occurrence (starting at lines 143 in the PDF).

# Reviewer 3

*The implementation is based on the SDSL library but important details, i.e. which SA sampling strategy was used, are missing in the current version of the article.*

We discussed the sampling distance in the "Evaluation procedure" section. However the discussion was focused on the simulated datasets. Now we also comment on what the sampling strategy involves for the real dataset.

*It is also unclear why the author opted for **rrr_vector** to represent the bit vectors. It is expected that **sd_vector** is superior to **rrr_vector** for long labels.*

This is a very good point and we agree with the reviewer that sd_vector is more suitable with low proportion of 1s ($< 10\,\%$). We added a discussion on that matter at the end of the "Conclusions" section. However we do not expect the $B_A$ or $B_D$ bit vector to dominate the space consumption of our index but rather the FM-index and the wavelet tree. Therefore the gain in space consumption would be only small.

# Other modifications

For the sake of clarity we introduced a new table (Table 1 in the manuscript) to summarize the time complexities of the queries for the two indexes we introduce as well as the baseline solution.

We also corrected several typos as pointed out by the reviewers. We would like to thank them again for their careful reading of the manuscript.

The authors